# On the Oral Microbiome of Oral Potentially Malignant and Malignant Disorders: Dysbiosis, Loss of Diversity, and Pathogens Enrichment

**DOI:** 10.3390/ijms24043466

**Published:** 2023-02-09

**Authors:** Alejandro Herreros-Pomares, David Hervás, Leticia Bagan-Debón, Eloísa Jantus-Lewintre, Concepción Gimeno-Cardona, José Bagan

**Affiliations:** 1Department of Biotechnology, Universitat Politècnica de València, 46022 Valencia, Spain; 2Centro de Investigación Biomédica en Red Cáncer (CIBERONC), 28029 Madrid, Spain; 3Department of Applied Statistics and Operational Research, and Quality, Universitat Politècnica de València, 46022 Valencia, Spain; 4Medicina Oral Unit, Stomatology Department, Valencia University, 46010 Valencia, Spain; 5Department of Microbiology, Hospital General Universitario de Valencia, 46014 Valencia, Spain; 6Department of Stomatology and Maxillofacial Surgery, Hospital General Universitario de Valencia, 46014 Valencia, Spain; 7Precancer and Oral Cancer Research Group, Valencia University, 46010 Valencia, Spain

**Keywords:** oral leukoplakia, oral cancer, microbiota, head and neck cancer, pathogens, *Campylobacter*, 16S rRNA

## Abstract

The role of dysbiosis in the development and progression of oral potentially malignant disorders (OPMDs) remains largely unknown. Here, we aim to characterize and compare the oral microbiome of homogeneous leucoplakia (HL), proliferative verrucous leukoplakia (PVL), oral squamous cell carcinoma (OSCC), and OSCC preceded by PVL (PVL-OSCC). Fifty oral biopsies from HL (*n* = 9), PVL (*n* = 12), OSCC (*n* = 10), PVL-OSCC (*n* = 8), and healthy (*n* = 11) donors were obtained. The sequence of the V3–V4 region of the 16S rRNA gene was used to analyze the composition and diversity of bacterial populations. In the cancer patients, the number of observed amplicon sequence variants (ASVs) was lower and *Fusobacteriota* constituted more than 30% of the microbiome. PVL and PVL-OSCC patients had a higher abundance of *Campilobacterota* and lower *Proteobacteria* than any other group analyzed. A penalized regression was performed to determine which species were able to distinguish groups. HL is enriched in *Streptococcus parasanguinis*, *Streptococcus salivarius*, *Fusobacterium periodonticum*, *Prevotella histicola*, *Porphyromonas pasteri*, and *Megasphaera micronuciformis*; PVL is enriched in *Prevotella salivae, Campylobacter concisus, Dialister pneumosintes*, and *Schaalia odontolytica*; OSCC is enriched in *Capnocytophaga leadbetteri, Capnocytophaga sputigena, Capnocytophaga gingivalis, Campylobacter showae, Metamycoplasma salivarium*, and *Prevotella nanceiensis*; and PVL-OSCC is enriched in *Lachnospiraceae bacterium, Selenomonas sputigena*, and *Prevotella shahii*. There is differential dysbiosis in patients suffering from OPMDs and cancer. To the best of our knowledge, this is the first study comparing the oral microbiome alterations in these groups; thus, additional studies are needed.

## 1. Introduction

According to a recent international consensus, oral potentially malignant disorders (OPMDs) are defined as any oral mucosal abnormality that is associated with an increased risk of occurrence of oral cancer [1]. Among them, oral leucoplakias are the most frequent form of OPMD and consist of a predominantly white plaque of uncertain risk, having excluded other known diseases that carry no increased risk for cancer [2]. In that sense, a recent systematic review and meta-analysis concluded that their potential for malignant transformation is 9.7% [3]. Furthermore, their location on the tongue and floor of the mouth, the existence of non-homogeneous clinical forms [4], and the histological findings of high-grade epithelial dysplasia [3] contribute to a higher risk of cancer development. Among oral leucoplakias, the proliferative verrucous leukoplakia (PVL) is a low-frequency, progressive, persistent, and irreversible subtype with distinctive clinical and evolutionary characteristics [5,6]. PVL is frequently located on the gingiva, has a much higher potential for malignant transformation (ranging from 43.87 to 71.4%), and shows an increased tendency to develop second primary tumors, with a very high recurrence rate after various treatments [7].

The morphological and cytological changes of OPMDs are similar to those of early invasive oral squamous cell carcinomas (OSCCs) and some of the chromosomal, genomic, and molecular alterations detected in them are also found in OPMDs [8]. Unfortunately, the literature is sparse about the microbiome associated with OPMDs and the potential role of microbial dysbiosis in OPMDs remains largely unknown.

The mouth, along with the aerodigestive tract, harbors the second most abundant microbiota in the human body. As of December 2022, the expanded Human Oral Microbiome Database (eHOMD) includes information on 774 bacterial species, 74% of them cultivable and 26% belonging to uncultivated phylotypes [9]. These oral microorganisms have a direct impact on their hosts, ranging from metabolic reactions to immune responses. Indeed, oral microbial dysbiosis has been reported in many diseases, including diabetes, bacteremia, endocarditis, cancer, autoimmune disease, and atherosclerosis, making it important to recognize the diversity of the oral microbiota and how it changes under altered conditions to determine its potential role in disease [10,11,12,13,14]. Dysbiotic oral microflora has already been associated with chronic periodontal disease [15,16] and oral cancer [17] and for PVL, a controversy exists around the possible implication of some pathogens, such as human papillomavirus (HPV) 16 [18,19,20,21]. In this regard, a recent study published by our group comparing PVL patients with healthy controls revealed that oral dysbiosis is a common state in PVL patients, who exhibited a general loss of diversity and enrichment of some protumorigenic pathogens, such as *Campylobacter jejuni* [22]. The aim of our new study, following the same line of research as the previous one, is to analyze the microbiota differences between homogeneous leukoplakia (HL) and PVL, as well as with two groups of cancers, one of them after the evolution of PVL.

## 2. Results

### 2.1. Participant Characteristics and Sequencing Data Summary

The clinical profile and clinicopathological information of the healthy donors (group I), the patients with HL (group II), PVL (group III), OSCC (group IV), and PVL-OSCC (group V) are shown in Appendix A. The mean patient age was 70.31 years, 52% were females, and 32% were smokers. No significant differences in clinicopathological characteristics were found between groups. Five patients with non-neoplastic lesions (HL and PVL) did not have dysplasia, twelve showed mild dysplasia, three presented moderate dysplasia, and one PVL case had severe dysplasia. In addition, PVL-OSCC patients tended to show a lower frequency of poorly differentiated tumors (G0/1: 50% vs. 75%), perineural infiltrations (25% vs. 70%), presence of lymph node metastasis in the neck (12.5% vs. 30%), and stage IV tumors (37.5% vs. 60%) compared to other OSCC.

Sequencing of oral samples and filtering for sequence quality resulted in a total of 2,435,181 effective reads, with a median of 44,927 reads per sample (range 16,160–110,317). Rarefaction curves (number of reads vs. alpha diversity) began to level off for most of the samples, indicating that samples were sequenced to a sufficient depth such that a complete microbiome profile was likely captured for most samples (Figure 1a). A total of 14 phyla, 24 classes, 62 orders, 102 families, and 189 genera were identified for *Bacteria*. Phylum-level classification of the bacterial community was also identified (Figure 1b) using the feature prevalence, which is the number of samples in which an amplicon sequence variant (ASV) appears at least once. The bacterial community was heavily dominated by phylum *Fusobacteriota* (relative abundance >26.7%), *Firmicutes* (>21.0%), *Bacteroidota* (>19.3%), and *Proteobacteria* (>17.0%) (Figure 1b). Regarding *Archea*, low numbers of *Woesearchaeales* of *Nanoarchaeota* and *Aenigmarchaeota* were also identified.

### 2.2. Differences in Microbiome Diversity

To study the complexity of the microbiota community structure within the samples, we determined alpha- and beta-diversity scores for all patients (Appendix A). We found that richness, as assessed by Chao1, was significantly higher for healthy donors (637.9 ± 306.8), PVL (652.7 ± 476.7), and HL (565.5 ± 295.8) than for OSCC (284.2 ± 166.9) and PVL-OSCC (320.5 ± 244.0) patients (*p* = 0.031) (Figure 1c). No significant differences were found in terms of diversity (Shannon–Wiener) or dominance (Simpson and Inverse Simpson). Regarding similarity, beta diversity revealed that the microbiome from OSCC and PVL-OSCC patients is more homogeneous than that from control, HL, and PVL patients (*p* = 2.95 × 10^−5^, Figure 1c).

### 2.3. Differences in Microbiome Composition

Krona plots representing an overview of the microbiome of all the samples included in the study are available in Appendix A. Additionally, a summarized description of the microbiome composition for each group of patients is provided in Table 1. The most abundant phyla found for all the groups were *Fusobacteriota*, *Bacteroidota*, *Proteobacteria*, and *Firmicutes* (Figure 2a). However, relative abundances significantly varied among groups. Thus, *Fusobacteriota* constituted 34% and 30% of the sequences detected in OSCC and PVL-OSCC patients, respectively, in contrast to the relative abundance of this phylum for healthy donors (24%), HL (25%), and PVL (19%). Of note, the relative abundance of *Campilobacterota* was higher in PVL (11%) and PVL-OSCC (14%) patients than in any other group analyzed. On the contrary, the relative abundance of *Proteobacteria* was lower in PVL (11%) and PVL-OSCC (11%) patients than in any other group (20% in controls, 25% in HL, and 19% in OSCC).

At the genus level, the most frequently detected genera in controls were *Haemophilus* (15%), *Fusobacterium* (14%), *Leptotrichia* (10%), and *Streptococcus* (10%) (Figure 2b). In contrast, *Streptococcus* (19%), *Fusobacterium* (19%), and *Haemophilus* (19%) were the most frequently registered genera in HL, and *Fusobacterium* (25%) and *Capnocytophaga* (13%) in OSCC. For PVL patients, *Streptococcus* (13%), *Fusobacterium* (14%), and *Campylobacter* (11%) were the genera with the highest relative abundance, whereas these were *Fusobacterium* (26%), *Campylobacter* (14%), and *Treponema* (10%) in the case of PVL-OSCC patients.

### 2.4. Community Structure Reveals Differently Abundant ASVs in Oral Disorders

In addition to the differences found in microbiome diversity and composition, specific ASVs were identified that exhibited differences in abundance between groups. Before further analysis, ASVs were filtered to remove microorganisms with very low counts (<100 reads) across all libraries since they provide little evidence for differential distribution analysis. After filtering, five genera were found to be exclusively present in healthy donors, whereas two were exclusive of PVL, four of OSCC, and four of PVL-OSCC (Figure 3a). Of note, no genera were found to be exclusive of HL, one was shared among OPMDs, and two were shared between OSCC and PVL-OSCC.

Afterward, a penalized regression was performed to determine which genera were able to distinguish groups, finding that 30 genera were differentially distributed among samples (Appendix A). Control samples were enriched in *Fretibacterium, Tannerella, Kingella, Corynebacterium, Lautropia, Cutibacterium, Saccharimonadaceae*, and *Pseudoramibacter*, whereas HL was enriched in *Rothia*, *Streptococcus*, *Megasphaera*, and *Mogibacterium*. PVL had a higher abundance of *Granulicatella, Gemella, Eubacterium, Actinomyces* and *Deep Sea Euryarchaeotic Group (DSEG)*; OSCC were enriched in *Capnocytophaga*, *Fusobacterium*, *Leptotrichia*, *Neisseria, Bergeyella, Mycoplasma, Johnsonella* and *Staphylococcus*; and PVL-OSCC in *Selenomonas, Catonella* and *Defluviitaleaceae UCG−011* (Figure 3b).

At the species level, seven microorganisms were found to be exclusively present in control samples, whereas three were exclusive of PVL, one of OSCC, and six of PVL-OSCC (Figure 4a). Again, no species were found to be exclusive of HL, three were shared between HL and PVL, and three were shared between OSCC and PVL-OSCC. In this case, the elastic net multinomial regression revealed that samples from healthy donors were enriched in *Campylobacter gracilis, Filifactor alocis, Phocaeicola abscessus*, and *Treponema maltophilum*; HL had more *Streptococcus parasanguinis*, *Streptococcus salivarius*, *Fusobacterium periodonticum*, *Prevotella histicola*, *Porphyromonas pasteri*, and *Megasphaera micronuciformis*; PVL were enriched in *Prevotella salivae*, *Campylobacter concisus*, *Dialister pneumosintes*, and *Schaalia odontolytica*; OSCC had a higher abundance of *Capnocytophaga leadbetteri*, *Capnocytophaga sputigena*, *Capnocytophaga gingivalis*, *Campylobacter showae*, *Metamycoplasma salivarium*, and *Prevotella nanceiensis*; and PVL-OSCC were enriched in *Lachnospiraceae bacterium*, *Selenomonas sputigena* and *Prevotella shahii* (Appendix A, Figure 4b).

## 3. Discussion

Microbiome studies, motivated by the availability of high-throughput technologies, have exhibited how the disturbance of the microbiota is associated with a great number of human diseases [23]. To date, the vast majority of studies have been performed on the gut, which constitutes the body niche where most of the commensal microorganisms reside. Although it has been less studied, oral microbiome dysbiosis seems to be linked to oral cancer and other oral diseases through several mechanisms, including the direct metabolism of chemical carcinogens and general inflammatory effects [24]. The oral cavity is home to more than 700 microbial species, including commensal and opportunistic bacterium, fungi, and viruses, however, there are still few studies addressing the influence of changes in the oral microbiome in the development and progression of OPMDs.

In this study, we characterized oral microbial communities of patients with HL, PVL, OSCC or OSCC preceded by PVL (PVL-OSCC). In terms of overall diversity, the average number of different species detected was lower in OSCC and PVL-OSCC than in disease-free, age-matched controls, but not in HL and PVL patients. This decrease in oral microbial diversity has been previously reported in several forms of head and neck cancers, including oral, esophageal, and nasopharyngeal carcinomas [25,26,27]. As with other studies comparing disease versus healthy microbiome, it is not possible to say whether the microbial alterations found are the cause or the consequence of the disease, however, the absence of this reduction in premalignant lesions could indicate that the decrease of the oral microbial diversity detected in cancer is a consequence of the disease. Nevertheless, reduction in oral microbial diversity has been reported in other oral diseases such as caries, recurrent aphthous stomatitis, or oral lichen planus, thus caution should be exerted when interpreting these findings.

At the phylum level, *Firmicutes*, *Bacteroidota*, *Proteobacteria*, *Actinobacteriota, Spirochaetota*, and *Fusobacteriota* have been reported to constitute 96% of the total oral bacteria [28], which aligns with the most abundant phyla found for all patients in our study. However, the abundance of *Campylobacterota* was >10% in PVL and PVL-OSCC in our study. Previous reports have shown the association between oral *Campylobacter* infections and increased risk of inflammatory bowel disease [29], esophageal [30], and oral [31] cancers. In addition, increased levels of *Campylobacterota* have been reported in oral leukoplakia (OL) [32] and a previous study performed by our group concluded that the microbiome of PVL patients is significantly enriched in *Campylobacter jejuni* [22]. Importantly, it is known that a *Campylobacter*-derived genotoxin, called cytolethal distending toxin (CDT), induces DNA double-strand breaks and facilitates colorectal tumorigenesis [33]. These findings, together with our data, suggest that *Campylobacter* species might be associated with PVL development and oral carcinogenesis via the induction of DNA damage.

At the genus level, the main components of the oral microbiome according to Bik et al. are *Actinomyces, Atopobium, Corynebacterium, Rothia* of *Actinobacteria*; *Bergeyella*, *Capnocytophaga*, *Prevotella* of *Bacteroidetes, Granulicatella*, *Streptococcus*, *Veillonella* of *Firmicutes*, *Fusobacterium*, *Campylobacter*, *Cardiobacterium*, *Haemophilus*, *Neisseria* of *Proteobacteria*, and TM7 [34]. However, differences can be found between studies, since many factors, including smoking habits, diet, and varying geographical and climatic conditions alter the oral microbiota, making comparisons difficult [35]. In our study, these bacteria were the most frequently detected genera in all the patients included, except for *Cardiobacterium, Actinomyces, Atopobium, Corynebacterium*, and TM7, which were less frequently detected than *Porphyromonas*, *Leptotrichia* or *Treponema*. Specifically, the presence of *Rothia*, *Streptococcus*, *Megasphaera,* and *Mogibacterium* was remarkable in HL, whereas PVL had a higher abundance of *Granulicatella, Gemella, Eubacterium, Actinomyces* and *Deep Sea Euryarchaeotic Group* (DSEG). Amer and colleagues compared swabs from OL to contralateral healthy sites and control samples and concluded that the microbiome of OL exhibits enrichment for *Fusobacterium*, *Leptotrichia*, *Campylobacter*, and *Rothia* species [36]. Moreover, another study comparing salivary samples from OL, OSCC and healthy controls reported that healthy controls could be distinguished from the former two disorders based on the abundance of *Megaspheara*, unclassified enterobacteria, Prevotella, *Porphyromonas, Rothia, Salmonella*, *Streptococcus*, and *Fusobacterium* [36]. Regarding oral cancers, in this study, OSCC were enriched in *Capnocytophaga*, *Fusobacterium*, *Leptotrichia*, *Neisseria, Bergeyella, Mycoplasma, Johnsonella,* and *Staphylococcus;* and PVL-OSCC in *Selenomonas, Catonella,* and *Defluviitaleaceae UCG−011*. Previous studies have reported the increased abundance of *Capnocytophaga*, *Fusobacterium*, *Leptotrichia*, *Staphylococcus, Selenomonas*, and *Catonella* in oral cancers [37,38,39,40]. In particular, Weiwen and collaborators reported that *Capnocytophaga* species are potential tumor promotors in oral cancer [37] and Saxena et al. reported that *Capnocytophaga* and *Fusobacterium* are differentially abundant in OSCC-associated microbiomes and can be considered as potential microbiome marker genera for oral cancer [39]. In addition, *Catonella* species are considered periodontitis-associated bacteria that may be related to primary endodontic infections [41,42].

In addition to the differences exposed in overall microbiome profiles, we also identified several species that were specifically over-represented in every oral disease. HL had a higher abundance of the protumorigenic pathogens *Prevotella histicola* and *Streptococcus parasanguinis*. *P. histicola* has been reported to produce acetaldehyde [43], whereas a study comparing the oral microbiota in tumor and non-tumor tissues of OSCC patients concluded that *S. parasanguinis* was highly associated with the tumor site [44]. The relative abundance of *S. parasanguinis* was also significantly higher in tongue/pharyngeal cancer patients [45]. Importantly, the relative abundance of the oral probiotic *Streptococcus salivarius* and *Porphyromonas pasteri* were also higher in HL. *S. salivarius* has been extensively reported as a probiotic microbe [46], whereas *P. pasteri* has been inversely associated with OSCC progression [47]. In contrast to HL, the oral microbiome of PVL was enriched in *Campylobacter concisus*, *Prevotella salivae*, and *Dialister pneumosintes*. Elevated levels of *C. concisus* have been associated with severe dysplasia in patients with OL, and a case-control study involving 25 OSCC cases and 27 fibroepithelial polyps (FEP) concluded that *C. concisus* was more abundant in OSCC than in FEP [48]. Additionally, this case-control study reported the enrichment of *P. salivae* in OSCC samples compared to FEP ones [48], whereas the study conducted by Coker and col. concluded that *D. pneumosintes* had significant centralities in the gastric cancer ecological network [49].

In OSCC, *Capnocytophaga leadbetteri*, *Capnocytophaga sputigena*, and *Capnocytophaga gingivalis* had a higher abundance. These three have been previously reported to be increased in OSCC patients, suggesting a potential association between these bacteria and OSCC [42,50,51]. Indeed, *C. sputigena* was suggested to cause invasive gingival disease with hyperplasia in immunocompromised patients [52] and *C. sputigena* bacteremia, most likely induced by an oral pathology, occurred in several patients with acute leukemia [53]. *C. gingivalis* is also considered a potential tumor promotor in oral cancer. A high salivary count of *C. gingivalis* has been suggested as a diagnostic indicator of OSCC [51], and its supernatant was found to induce epithelial to mesenchymal transition (EMT), causing OSCC cells to acquire highly invasive and metastatic properties [37]. *Metamycoplasma salivarium* and *Prevotella nanceiensis* were also elevated in OSCC. *M. salivarium* was considered a non-pathogenic commensal at first, however, its detection in the epithelial cells of OL and oral lichen planus and as a dominant colonizer of Fanconi anemia-associated oral carcinoma has cast doubt on this [54,55]. On the other hand, the higher abundance of *P. nanceiensis* has been suggested to influence OSCC through inflammation [56]. In contrast to OSCC, PVL-OSCC were enriched in *Selenomonas sputigena, Lachnospiraceae bacterium*, and *Prevotella shahii*, and, to the best of our knowledge, it is the first time that these bacteria have been linked to oral cancers.

## 4. Materials and Methods

### 4.1. Patients and Tissue Samples

This study included 50 individuals who visited and were treated at the Stomatology and Maxillofacial Surgery Department of the General University Hospital of Valencia. Participants were distributed into five groups according to their oral disorder: group I consisted of nine patients with HL, group II included twelve patients with PVL, group III comprised ten OSCC patients, group IV was composed of eight patients with OSCC preceded by the evolution of PVL (PVL-OSCC), and group V included eleven healthy donors. For groups I to IV, two representative biopsies were taken from the same area of the lesions, including epithelium and the underlying connective tissue between 2017 and 2021. One of each pair of specimens was analyzed with the routine histopathological methods to ensure that the observed lesions met the histopathological criteria to establish the diagnosis together with the clinical data of each patient. PVL diagnosis was determined following the criteria provided by Villa et al. [6]. The other sample was used for the 16S sequencing. For the control group (group V), samples were obtained from healthy mucosa areas adjacent to the teeth (vestibular fundus). All tissue samples were frozen at −80 °C until their analysis.

### 4.2. DNA Extraction and 16S rDNA Gene Sequencing

Total DNA from clinical samples was extracted using the DNAeasy kit (QiaGen, Barcelona, Spain). Variable V3 and V4 regions of the 16S rDNA gene were amplified following the 16S Metagenomic Sequencing Library Preparation Illumina protocol (Cod. 15044223 Rev. A, Illumina, Inc., San Diego, CA, USA). The full-length primer sequences including Illumina adapter overhang nucleotide sequences were selected according to Klindworth et al. [57] as follows:

Forward primer: 5′-TCGTCGGCAGCGTCAGATGTGTATAAGAGACAGCCTACG-GGNGGCWGCAG-3′.

Reverse primer: 5′-GTCTCGTGGGCTCGGAGATGTGTATAAGAGACAGGACTA-CHVGGGTATCTAATCC-3′.

After 16S rDNA gene amplification, the multiplexing step was performed using Nextera XT Index Kit (FC-131-2001). A volume of 1 μL of the PCR product was run on a Bioanalyzer DNA 1000 chip to verify the size (~550 bp expected) on a Bioanalyzer 2100 (Agilent, Santa Clara, CA, USA). After size verification, libraries were sequenced using a 2 × 300 bp paired-end run (MiSeq Reagent kit v3, MS-102-3003) on an Illumina MiSeq Sequencer according to the manufacturer’s instructions.

### 4.3. Bioinformatic Analysis and Taxonomic Annotation

Sequencing data were demultiplexed using the Illumina bcl2fastq© program. Demultiplexed paired FASTQ sequences were processed using QIIME2 v2021.4. Quality control was carried out using the DADA2 pipeline incorporated into QIIME2 [58]. The DADA2 pipeline filtered out phiX reads, removed chimeric sequences, and assigned reads into Amplicon Sequence Variants (ASVs) [59]. Taxonomic affiliations were assigned using the Naive Bayesian classifier integrated into QIIME2 plugins. The SILVA v138 database was used for taxonomic annotation [60]. The taxonomic composition of the oral microbiota was generated by different levels: kingdom, phylum, class, order, family, genus, and species.

### 4.4. Statistical Analysis

Data were summarized using mean (standard deviation) and median (first and third quartiles) in the case of continuous variables, and absolute (relative) frequencies in the case of categorical variables. Unsupervised analyses were performed using principal coordinates analysis and hierarchical clustering. Chi-squared test and nonparametric tests (Mann–Whitney U or Kruskal–Wallis) were applied to evaluate associations between patient clinicopathological characteristics and microbiota composition. Compositional differences among groups were assessed by elastic net multinomial regression. The penalization factor for the elastic net analysis was selected using three-fold cross-validation. Statistical analyses were performed using R software v.4.2.2 (R Foundation for Statistical Computing, Vienna, Austria), and the R packages microbiome (version 1.18.0), pgirmess (version 2.0.0), glmnet (version 4.1-4) and clickR (version 0.9.27).

## 5. Conclusions

Using a 16S rRNA gene sequencing-based approach, oral microbial dysbiosis was found in patients with HL, PVL, OSCC, and PVL-OSCC. Loss of diversity was found in oral cancers but not in premalignant lesions, which could indicate that the decrease of the oral microbial diversity detected in cancer is a consequence of the disease. In addition, the relative abundance of *Campylobacterota* was higher in PVL and PVL-OSCC, which may constitute an important risk factor for PVL development and progression. Oral diseases could be distinguished by the abundance of certain species. A better understanding of the role of the oral microbiome in OPMDs and cancer could direct to novel non-invasive diagnostic and prognostic options, as well as to more personalized treatments and microbiome-targeted therapeutic interventions.

## Figures and Tables

**Figure 1 ijms-24-03466-f001:**
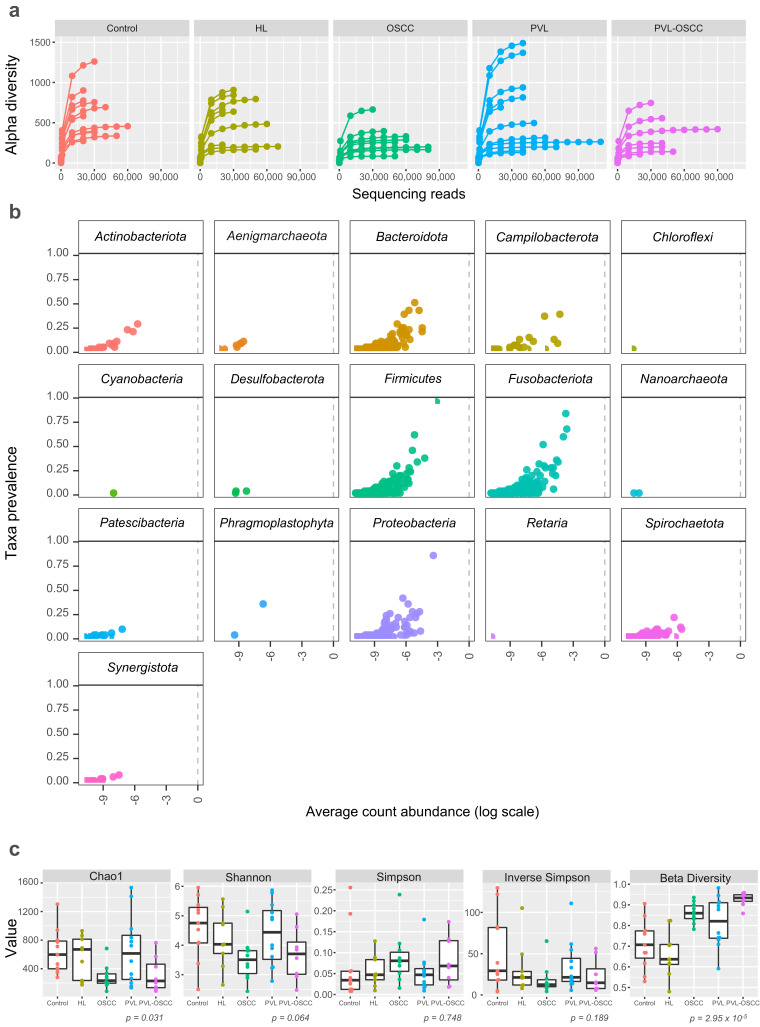
Microbiome diversity for oral samples collected from healthy controls and HL, PVL, OSCC, and PVL-OSCC patients. (**a**) Rarefaction curves (alpha diversity versus sequencing library size). (**b**) Prevalence plot (taxa prevalence versus average count abundance). Each point corresponds to a different amplicon sequence variant. (**c**) Microbial richness (Chao1), diversity (Shannon), dominance (Simpson and Inverse Simpson), and similarity (beta divergence) of oral samples.

**Figure 2 ijms-24-03466-f002:**
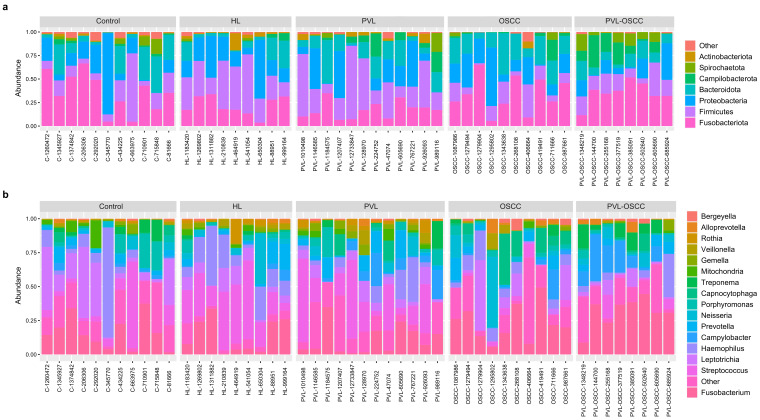
Stacked bar plot of the phylogenetic composition of the most common taxa at the phylum (**a**) and genus (**b**) levels.

**Figure 3 ijms-24-03466-f003:**
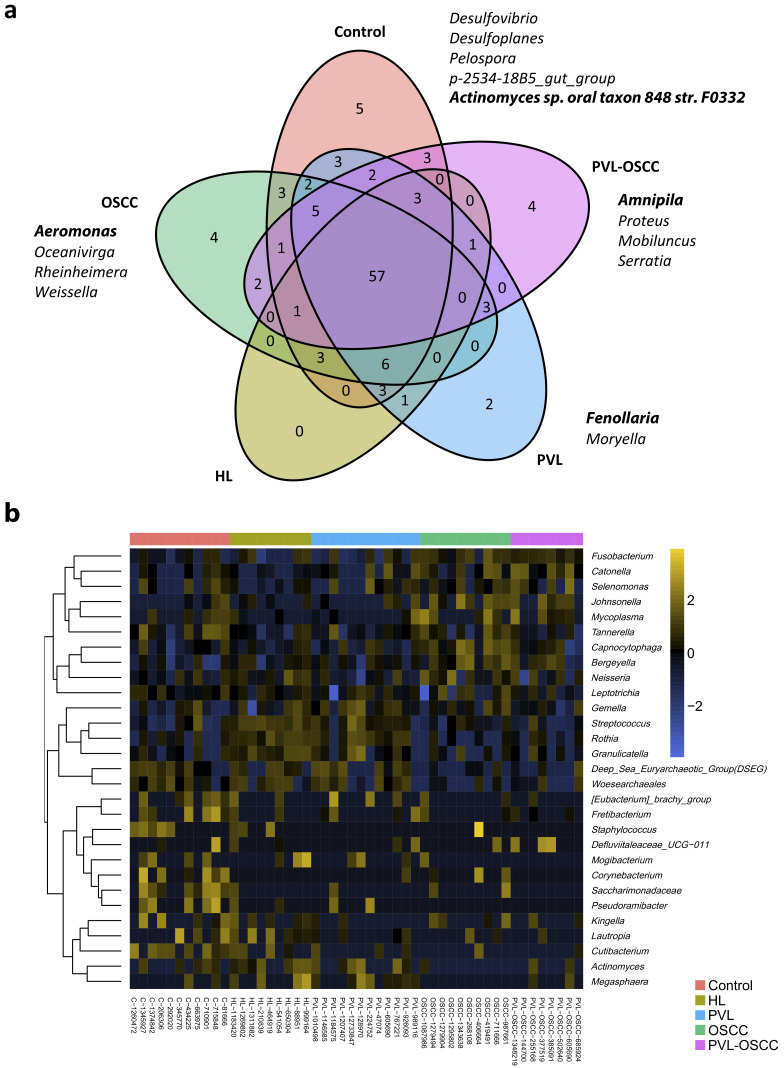
Differently abundant genera between samples from healthy donors, HL, PVL, OSCC, and PVL-OSCC patients. (**a**) Venn diagram summarizing the distribution of genera along groups. (**b**) Clustering of the differentially distributed genera along samples as selected by elastic net analysis.

**Figure 4 ijms-24-03466-f004:**
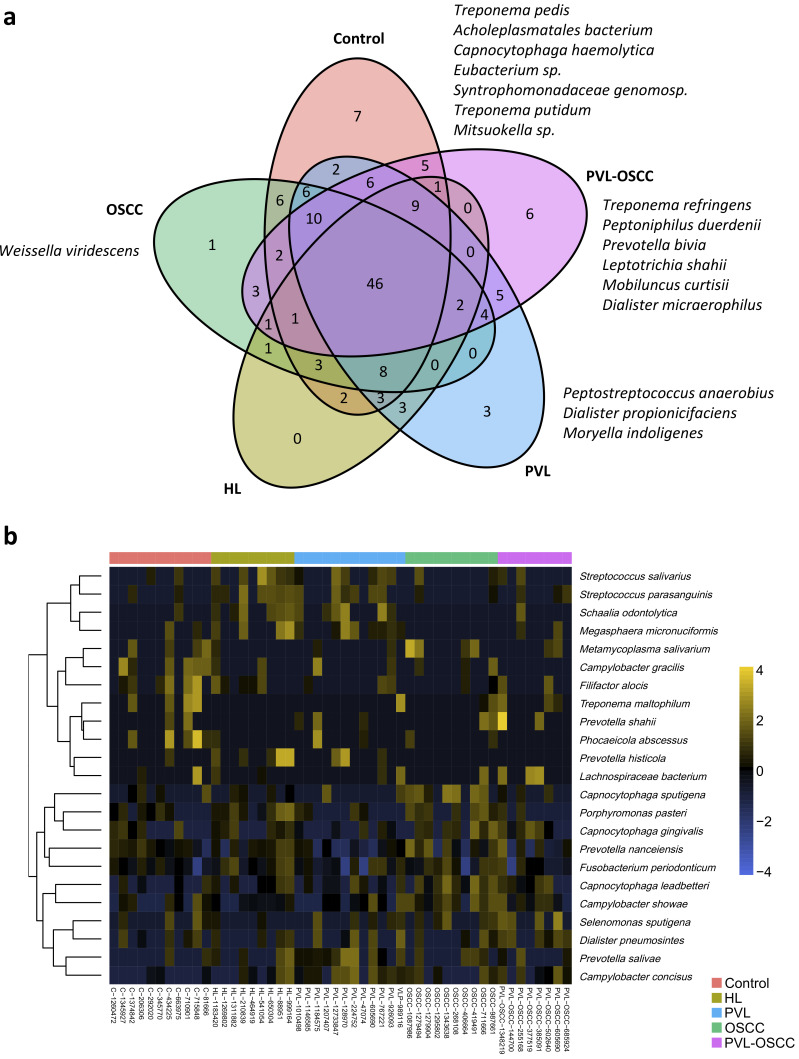
Differently abundant species between samples from healthy donors, HL, PVL, OSCC, and PVL-OSCC patients. (**a**) Venn diagram summarizing the distribution of species along groups. (**b**) Clustering of the differentially distributed species along samples as selected by elastic net analysis.

**Table 1 ijms-24-03466-t001:** Phylogenetic composition of the most common taxa by group. Abundances at the phylum and genus levels are presented as a percentage of the total microbiome. Abundances < 1% are presented as empty cells. Asterisks indicate *p*  <  0.05.

Phylum	Genus	Control	HL	PVL	OSCC	PVL-OSCC	*p*-Value
*Fusobacteriota*		24%	25%	19%	34%	30%	0.021 *
	*Fusobacterium*	14%	19%	14%	25%	26%	0.006 *
	*Leptotrichia*	10%	5%	5%	9%	4%	0.056
*Bacteroidota*		21%	14%	20%	25%	13%	0.195
	*Porphyromonas*	9%	3%	6%	3%	3%	0.433
	*Prevotella*	3%	8%	9%	5%	4%	0.689
	*Tannerella*	3%					0.002 *
	*Capnocytophaga*	2%		1%	13%	1%	0.012 *
	*Alloprevotella*		1%	1%	2%	2%	0.422
	*Bergeyella*				2%	1%	0.034 *
*Proteobacteria*		20%	25%	11%	19%	11%	0.708
	*Haemophilus*	15%	19%	7%	7%	4%	0.459
	*Neisseria*	2%	5%	1%		3%	0.518
	*Aggregatibacter*			1%	1%	1%	0.671
	*Pseudomonas*				2%		0.297
*Firmicutes*		19%	26%	28%	14%	21%	0.011 *
	*Streptococcus*	10%	19%	13%	3%	3%	0.002 *
	*Gemella*	1%	1%	2%			0.103
	*Dialister*	1%					0.398
	*Selenomonas*	1%				3%	0.236
	*Veillonella*		2%	2%	2%		0.287
	*Granulicatella*		1%	1%			0.182
	*Fenollaria*			1%			0.546
	*Catonella*			1%		4%	0.001 *
	*Mycoplasma*				3%	2%	0.229
*Spirochaetota*		5%		5%	1%	10%	0.020 *
	*Treponema*	5%		5%	1%	10%	0.022 *
*Campylobacterota*		2%	3%	11%	4%	14%	0.002 *
	*Campylobacter*	2%	3%	11%	4%	14%	0.003 *
*Synergistota*		1%					0.029 *
	*Fretibacterium*	1%					0.020 *
*Actinobacteriota*		1%	2%	2%			0.059
	*Rothia*		1%	1%			0.018 *
Others		7%	5%	4%	3%		--

HL, homogeneous leukoplakia; PVL, proliferative verrucous leukoplakia, OSCC, oral squamous cell carcinoma; PVL-OSCC, oral squamous cell carcinoma preceded by proliferative verrucous leukoplakia.

## Data Availability

Raw data have been deposited at the NCBI SRA archive with BioProject record PRJNA916491 and BioSample records from SAMN32471763 to SAMN32471812.The samples were managed and processed by the Biobank of the Hospital Universitario y Politécnico, authorised biobank (B.0000723) following the requirements of RD1716/2011.

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
