# Peer review of "On the Oral Microbiome of Oral Potentially Malignant and Malignant Disorders: Dysbiosis, Loss of Diversity, and Pathogens Enrichment"

_ijms, 2023, doi:10.3390/ijms24043466_

Round 1
Reviewer 1 Report
In the original article entitled “How is the oral microbiome of the oral potentially malignant and malignant disorders? Dysbiosis, loss of diversity and pathogens enrichment” by Herreros-Pomares et al (Manuscript ID: ijms-2190562), the authors have compared the oral microbiome alterations in patients suffering from OPMDs and cancer. It is an interesting scientific topic. However, there are several points that need to be addressed and the manuscript should be revised before it can be considered for acceptance.
Major comments
1. Table 1 should be incorporated into the section of supplemental results.
2. The statistical analyzed results (P value) should be incorporated into some Figures (e.g. Fig1 C) and Table 2.
3. The presented data in the Fig1 b is difficult to understand, the authors should provide more explanations here.
4. In the Fig 1, the authors should show the beta diversity data firstly, then the divergence.
5. The section of Conclusion is tedious. The authors should carefully reorganize these contents.
Minor comments
1. The alphabets in Figures are not consistent with that in text (e.g. Figure 1a vs. Figure1A).
2. In Figure 1C, the Shannon index represents bacterial diversity not just evenness.
3. The bacterial names in genus and species levels used in the text should be italicized.
4. Some abbreviations used in the text were not listed (Line 389).
Author Response
In the original article entitled “How is the oral microbiome of the oral potentially malignant and malignant disorders? Dysbiosis, loss of diversity and pathogens enrichment” by Herreros-Pomares et al (Manuscript ID: ijms-2190562), the authors have compared the oral microbiome alterations in patients suffering from OPMDs and cancer. It is an interesting scientific topic. However, there are several points that need to be addressed and the manuscript should be revised before it can be considered for acceptance. Major comments 1. Table 1 should be incorporated into the section of supplemental results. Response: According to the reviewer’s suggestion, the table has been incorporated into supplementary data. 2. The statistical analyzed results (P value) should be incorporated into some Figures (e.g. Fig1 C) and Table 2. Response: According to the reviewer’s proposal, we have included p-values in Figure 1 and Table 2. 3. The presented data in the Fig1 b is difficult to understand, the authors should provide more explanations here. Response: According to the suggestion of both reviewers, an explanatory sentence has been added to the Results section: “Phylum-level classification of the bacterial community was also identified (Figure 1b) using the feature prevalence, which is the number of samples in which an amplicon sequence variant (ASV) appeared at least once”. 4. In the Fig 1, the authors should show the beta diversity data firstly, then the divergence. Response: Indeed, the results shown in Fig 1 corresponded to beta diversity. We have changed beta divergence for beta diversity to prevent misunderstanding. 5. The section of Conclusion is tedious. The authors should carefully reorganize these contents. Response: The Conclusion section has been summarized as follows to make it clearer: “Using a 16S rRNA gene sequencing-based approach, oral microbial dysbiosis was found in patients with HL, PVL, OSCC, and PVL-OSCC. Loss of diversity was found in oral cancers, but not in premalignant lesions, which could indicate that the decrease of the oral microbial diversity detected in cancer is a consequence of the disease. In addition, the relative abundance of Campylobacterota was higher in PVL and PVL-OSCC, which may constitute an important risk factor for PVL development and progression. Oral diseases could be distinguished by the abundance of some species. Thus, a better understanding of the role of the oral microbiome in OPMDs and cancer could direct to novel non-invasive diagnostic and prognostic options, as well as to more personalized treatments and microbiome-targeted therapeutic interventions”. Minor comments 1. The alphabets in Figures are not consistent with that in text (e.g. Figure 1a vs. Figure1A). Response: We have adapted alphabets in the text to make them consistent with alphabets in the Figures. 2. In Figure 1C, the Shannon index represents bacterial diversity not just evenness. Response: We have changed evenness for diversity for the Shannon index in the text and figure caption: “No significant differences were found in terms of diversity (Shannon-Wiener)”. 3. The bacterial names in genus and species levels used in the text should be italicized. Response: We have italicized all the bacterial names as proposed by the reviewer. 4. Some abbreviations used in the text were not listed (Line 389). Response: All the abbreviations used in the text have been incorporated into the abbreviations list.Reviewer 2 Report
The manuscript by Herreros-Pomares et al., “How is the oral microbiome of the oral potentially malignant 2 and malignant disorders? Dysbiosis, loss of diversity and pathogens enrichment” aims to compare the oral microbiome alterations between healthy subjects and those suffering from cancer or oral potentially malignant disorders. Authors seemed to home in on the important aspects of the study, with an in-depth discussion of relevant findings. The manuscript overall was well-constructed, concise and stayed on topic. The one suggestion is to include a better description of the data presented in Figure 1, especially part b.
Author Response
The manuscript by Herreros-Pomares et al., “How is the oral microbiome of the oral potentially malignant 2 and malignant disorders? Dysbiosis, loss of diversity and pathogens enrichment” aims to compare the oral microbiome alterations between healthy subjects and those suffering from cancer or oral potentially malignant disorders. Authors seemed to home in on the important aspects of the study, with an in-depth discussion of relevant findings. The manuscript overall was well-constructed, concise and stayed on topic. The one suggestion is to include a better description of the data presented in Figure 1, especially part b. Response: According to the suggestion of both reviewers, an explanatory sentence has been added to the Results section: “Phylum-level classification of the bacterial community was also identified (Figure 1b) using the feature prevalence, which is the number of samples in which an amplicon sequence variant (ASV) appeared at least once”.Round 2
Reviewer 1 Report
1. The authors should provide a one revised version with the markers for easier review.
2. I think the authors performed the Kruskal-Wallis test allows to compare three or more groups in the manuscript. The authors just show the general p-value in the Fig1 C and Table 2. However, the reader cannot clearly know the statistical difference between these two groups. So, the authors should provide the information in the figures or table (e.g. using different letters).
3. Table2 → Table1 in the manuscript.
Author Response
Dear Editor and Reviewers,
We are grateful for your effort, the helpful comments received, and the opportunity to improve the quality and scientific level of this research. Our careful revision addressed all the suggestions made to the present study. Point-by-point responses to the comments are listed below.
Reviewer 1
- The authors should provide a one revised version with the markers for easier review.
Response: According to the reviewer’s suggestion, we include a version with Track changes activated and a “clean” one this time. Unfortunately, we cannot provide a version with Tracked changes for the first round of revision anymore.
- I think the authors performed the Kruskal-Wallis test allows to compare three or more groups in the manuscript. The authors just show the general p-value in the Fig1 C and Table 2. However, the reader cannot clearly know the statistical difference between these two groups. So, the authors should provide the information in the figures or table (e.g. using different letters).
Response: Unfortunately, the sample size of the study is limited (HL (N = 9), PVL (N = 12), OSCC (N = 10), PVL-OSCC (N = 8), and healthy (N = 11) donors). Performing paired comparisons require adjustment for multiple comparisons and after this adjustment, non-significant differences are found. Thus, compositional differences among groups were assessed by elastic net multinomial regression (Figures 3 and 4).
- Table2 → Table1 in the manuscript.
Response: Changes have been applied.